# Effect of Silicon Source (Fly Ash, Silica Dust, Gangue) on the Preparation of Porous Mullite Ceramics from Aluminum Dross

**DOI:** 10.3390/ma15207212

**Published:** 2022-10-16

**Authors:** Hong-Liang Yang, Zi-Shen Li, You-Dong Ding, Qi-Qi Ge, Yu-Juan Shi, Lan Jiang

**Affiliations:** 1Key Laboratory for Ecological Metallurgy of Multimetallic Mineral, Ministry of Education, Northeastern University, Shenyang 110819, China; 2School of Metallurgy, Northeastern University, Shenyang 110819, China; 3Northeastern University Engineering & Research Institute Co., Ltd., Shenyang 110819, China

**Keywords:** aluminum dross, mullite, porous ceramics, silicon source

## Abstract

Aluminum dross (AD) is a waste product produced during aluminum processing and can be used to prepare mullite ceramic materials. However, the research on the preparation of mullite porous ceramics entirely from solid waste is still in the development stage. In this paper, porous mullite ceramics were successfully fabricated using a solid-phase sintering process with AD and different silicon sources (fly ash, silica dust, and gangue) as raw materials. The bulk density, apparent porosity, and compressive strength of the specimens were obtained, and the phase compositions and microstructures of the sintered specimens were measured using XRD and SEM, respectively. The average activation energy of the phase transition of fly ash, silica dust, and gangue as silicon sources were 984 kJ/mol, 1113 kJ/mol, and 741 kJ/mol, respectively. The microstructures of the mullite in the specimens were prisms, random aggregates, and needle-shaped, respectively. The formation of needle-shaped mullite combined with the substrate enhanced the mechanical strength of the porous mullite ceramics. The apparent porosity, density, and compressive strength of the specimens with gangue as the silicon source were 33.13%, 1.98 g/cm^3^, and 147.84 MPa, respectively, when sintered at 1300 °C for 2 h.

## 1. Introduction

Aluminum dross (AD) is an industrial solid waste produced in the process of Al metal smelting, when the molten metal surface is in contact with the air [1,2,3]. With the expansion of aluminum production scale, the output of AD is increasing rapidly. The accumulated AD not only occupies land but also cause economic and environmental problems [4,5]. AlN in AD will hydrolyze with water, even at room temperature, producing NH_3_ and polluting the environment, and the accumulation of AD will cause the leaching of salts (KCl, NaCl, NaF, et al.) and gradually destroy the soil environment [6,7,8]. In our previous works [9,10], AD was treated using a hydrometallurgy process, and the AlN and soluble salts in AD were removed. The main component of the AD was rich in alumina, which is a renewable resource for the preparation of mullite materials.

Mullite porous ceramics is a widely studied ceramic [11], because it exhibits useful properties such as high temperature resistance, oxidation resistance, low thermal conductivity, small expansion coefficient, good thermal shock resistance, and high creep resistance [12,13,14]. It has been widely used in metallurgy, glass, ceramics, chemistry, electricity, national defense, gas, and cement, as well as other industrial fields, because of its distinctive properties [14,15].

Mullite is a stable binary compound in Al_2_O_3_-SiO_2_ system, which is a solid solution composed of 3Al_2_O_3_-2SiO_2_ and 2Al_2_O_3_-SiO_2_. However, natural mullite ore is rare, owing to its strict formation conditions. The main raw materials for synthesizing mullite are industrial alumina [16,17,18], bauxite [14], and kaolin [19,20]. However, these materials are obtained from non-renewable resources or have a high cost, and do not conform to the national energy conservation, emission reduction, and sustainable development concept. Therefore, in recent years, the preparation of mullite ceramics from solid wastes, such as AD [21,22], fly ash [23], silica dust [22], and gangue [24], and replacing industrial raw materials completely, has become a new research hotspot, which is beneficial for the reuse of solid waste resources and the reduction of production costs. The main components of these solid wastes are Al_2_O_3_ or SiO_2_, and mullite ceramics can be successfully sintered at 1300–1500 °C. For example, Ibarra, C.M.N. [25] prepared mullite-zirconia composites at 1500 °C using AD and zirconia as raw materials. Ma, B.Y. [26] reported that the porous mullite ceramics were successfully fabricated with a fly ash and bauxite via reaction synthesis process at 1450–1550 °C.

Apparently, it is feasible and advisable to synthesize mullite porous ceramics from solid wastes [27,28,29,30,31]. However, there are still some problems with the preparation of mullite from AD. In the process of sintering, AlN and salts in AD will seriously affect the properties of mullite ceramics, and there has been little research on the preparation of mullite ceramics completely from solid wastes. In addition, the sintering temperature is usually higher than 1400 °C, which causes a high energy consumption.

In the present work, in order to make better use of AD and realize the comprehensive utilization of solid waste, three different silicon sources (fly ash, silica dust, and gangue) were added in the process of forming mullite at 900–1300 °C, and a range of porous mullite ceramics were prepared using a solid-phase sintering process. The work was mainly focused on the effects of the silicon source and sintering temperature on the properties of the fabricated ceramics, especially the phase transition and kinetics calculation.

## 2. Experimental

### 2.1. Materials

The chemical compositions of the AD (from Inner Mongolia Heng-sheng Environmental Protection Technology Co., Ltd. Tongliao, China. D_50_ = 22.74 μm), fly ash (from Boulder Mining Co., Ltd., Shijiazhuang, China. D_50_ = 25.73 μm), silica dust (from Aotai Mineral Products Co., Ltd., Shijiazhuang, China. D_50_ = 25.86 μm), and gangue (from Yunshi Building Materials Co., Ltd., Shijiazhuang, China. D_50_ = 8.05 μm) are shown in Table 1. An X-ray diffraction (XRD) analysis of the raw materials is shown in Figure 1 and Figure 2 shows the microstructure of the raw materials.

### 2.2. Experimental Procedure

According to the theoretical molar ratio of mullite (Al_2_O_3_:SiO_2_ = 3:2), the corresponding weight of raw materials (in Table 2) was weighed, put in a high-energy planetary ball mixer for 2 h, and then put into a sample bag for 6 h. After that, the powders were pressed into several batches of cylinder green bodies (φ25 mm × 10 mm) under a pressure of 5 MPa. All green bodies were then dried at 105 °C and sintered in air atmosphere at 900–1300 °C for 2 h in a resistance furnace, with a heating rate of 5 K/min. Finally, the specimens were cooled to room temperature inside the furnace.

### 2.3. Analytical and Characterization Methods

The XRD patterns of the raw materials and sintered specimens were analyzed using an X-ray diffractometer (D8 ADVANCE, Bruker AXS Co., Ltd., Karlsruhe, Germany) with Cu-Kα (40 kV, 80 mA); the scanning range was 10~90°, and the scanning speed was 10°/min. The XRF analysis of the raw materials was conducted using an X-ray fluorescence spectrometer (ZSX Primus II, Neo-Confucianism Co., Ltd., Tokyo, Japan). The optical tube voltage was 60 kV, and the current was 150 mA. Scanning electron microscopy (SEM) and energy dispersive spectroscopy (EDS) were performed with a field emission scanning electron microscope (Quanta 250FEG, FEI Co., Ltd., Brno, Czech Republic). The acceleration voltage was 30 kV. The particle size distribution of the raw materials was determined using a laser particle size analyzer (Bettersize 2000, Bettersize Instruments Co., Ltd., Dandong, China). Thermogravimetric (TG) and differential thermal analysis (DTA) were processed using an HQT-4 integrated thermal analyzer (Beijing HENVEN Company in China). The TG-DTA were conducted at heating rates of 10, 20, 30, and 40 K/min in nitrogen atmosphere, with a heating range from room temperature to 1475 K. A universal material testing machine (CMTH-1, Meitesi Testing Machine Factory, Tianjin, China) was used to measure the compressive strength of the porous ceramics, and the loading rate was 0.5 mm/min. The apparent porosity and bulk density of the porous ceramics were measured according to the Chinese standard GB/T 1966–1996.

## 3. Results and Discussion

### 3.1. Effects of Silicon Source on Phase Compositions

The XRD patterns of specimens with different silicon sources after sintering at 900–1300 °C for 2 h are presented in Figure 3.

The major crystalline phases detected were Al_2_O_3_(corundum), 3Al_2_O_3_·2SiO_2_(mullite), and MgAl_2_O_4_(spinel).

Compositional differences in silicon sources can cause differences in phase composition, such as the anorthite (Ca(Al_2_Si_2_O_8_), in Figure 1b) contained in fly ash. The formation temperatures of the mullite phase in the porous mullite ceramic prepared with fly ash, silica dust, and gangue were 1200 °C, 1200 °C, and 1100 °C, respectively. In particular, kaolinite, which is mainly contained in gangue, will first form mullite nuclei after being calcined to remove crystal water [32]. The reaction [25,33,34,35] is shown in Equations (1)–(4).

When the temperature is 450~800 °C
(1)$${2\mathrm{SiO}}_{2}\cdot {\mathrm{Al}}_{2}O_{3}\cdot {2H}_{2}{O=2\mathrm{SiO}}_{2}\cdot {\mathrm{Al}}_{2}O_{3}{+2H}_{2}O$$

When the temperature is 800~980 °C
(2)$${2\mathrm{SiO}}_{2}\cdot {\mathrm{Al}}_{2}O_{3}{=\mathrm{SiAl}}_{2}O_{5}{+\mathrm{SiO}}_{2}$$
or
(3)$${2\mathrm{SiO}}_{2}\cdot {\mathrm{Al}}_{2}O_{3}{=\mathrm{Al}}_{2}O_{3}{+2\mathrm{SiO}}_{2}$$

When the temperature >1100 °C
(4)$${\mathrm{SiAl}}_{2}O_{5}{+\mathrm{SiO}}_{2}=1/3\left({3\mathrm{Al}}_{2}O_{3}\cdot {2\mathrm{SiO}}_{2}\right){+4/3\mathrm{SiO}}_{2}$$

In addition, the formation temperature of the mullite phase is lower than the synthesis temperature required by the solid-phase sintering process reported in the literature [32], which is mainly due to the existence of AlOOH and Al(OH)_3_ in the AD promoting the high temperature crystallization of mullite [36,37].

### 3.2. Effects of Silicon Source on Morphologies

Figure 4 and Figure 5 show SEM images and the EDS spectrums of the specimens with different silicon sources sintered at 1200 °C and 1300 °C for 2 h, respectively. The crystal structures of mullite formed in the specimens prepared with fly ash, silica dust, and gangue as silicon sources are prisms (Figure 4a), random aggregates (Figure 4b), and needle-shaped (Figure 4c) when sintering at 1200 °C for 2 h, respectively. Impurities in gangue led to the formation of silicate liquid phase; and according to the L-S mechanism [20,38,39], the existence of liquid phase can promote the development of needle-shaped mullite. When the sintering temperature reached 1300 °C, the specimens with different silicon sources all showed different degrees of melting state, the glass phase increased significantly, and the mullite phase and the substrate became fused together.

EDS analyses of the points I-VI in Figure 4 and Figure 5 are shown in Table 3. The results in Table 3 show that Mg elements (points I and IV) were always present in the mullite structure of the specimens prepared with fly ash as the silicon source, indicating that the specimens were composed of a mullite–magnesium aluminum spinel complex. When silica dust was the silicon source, the Si element content at 1200 °C was 20.55% (point II), which is much larger than the Si element content in the theoretical value of mullite, indicating a silicon-rich state, which is the main reason why it is difficult to form crystals. In the samples prepared with gangue as the silicon source, only Al, Si, and O elements were in the needle-shaped mullite (point III).

### 3.3. Effects of Silicon Source on Phase Transition Kinetics

Through kinetic calculation of the thermal analysis data, the process and mechanism of chemical reaction of substances could be understood, and kinetic parameters such as reaction rate could be determined. Commonly used non-isothermal kinetic analysis methods include the Kissinger method [37] (Equation (5)) and Ozawa method [40] (Equation (6)). The activation energy $E_{a}$ of the reaction could be calculated according to the slope of the linear fitting curve of each equation.
(5)$$\ln\left(\frac{T_{P}^{2}}{\alpha }\right)=\ln\left(\frac{E_{a}}{R}\right)+\frac{E_{a}}{{\mathrm{RT}}_{P}}-\ln\nu$$
(6)$$\ln\alpha =-\frac{E_{a}}{{\mathrm{RT}}_{P}}+C$$
where T_p_ was the absolute temperature at the exothermic peak of the DTA curve in K; α was the heating rate in °C·min^−1^; $E_{a}$ was the activation energy in KJ·mol^−1^; R was the pervasive gas constant, and ν was the frequency factor; C was constant.

The results of TG-DTA analysis of different silicon sources (20 K/min) are shown in Figure 6. The DTA curve in Figure 6 shows that the exothermic peak positions of the samples prepared with fly ash, silica fume and coal gangue as silicon sources were 1380 K, 1430 K and 1270 K, respectively. TG curves showed that the weight loss rates of the samples prepared with fly ash, silica dust, and gangue as silicon sources were 11.8%, 4.6%, and 14.1% at 1400 K, respectively. The main reasons for this difference were the different compositions and amounts of the different silicon sources added.

Figure 7 shows the exothermic peak of DTA curves of AD mixed with three silicon sources at different heating rates. With the increase of heating rate, the exothermic peaks were shifted to the high temperature region, due to thermal hysteresis.

According to the analysis results in Figure 7, the phase transition kinetics fitting results of AD mixed with the three silicon sources based on Kissinger (Equation (5)) and Ozawa (Equation (6)) method are shown in Figure 8 and Figure 9, respectively. It can be seen from the figures that the correlation coefficients R^2^ of the three samples based on different methods were all above 0.97, and the results of the relevant kinetic parameters are shown in Table 4.

The results in Table 4 showed that the average activation energy of the phase transition process of AD mixed with fly ash, silica dust, and gangue calculated using the Kissinger and Ozawa methods were 984 kJ/mol, 1113 kJ/mol, and 741 kJ/mol, respectively. The silicon sources had different effects on the activation energy of the phase transition process of mullite synthesis with AD; among which, the activation energy of AD mixed with gangue was the smallest. The reason for this difference was that the composition and impurity content of the silicon sources were different, and the thermal decomposition of kaolinite in gangue produced primary mullite, which could reduce the activation energy when forming the needle-shaped mullite.

### 3.4. Effects of Silicon Sources on Physical Properties

The physical properties of specimens with different silicon sources sintered at 900–1300 °C for 2 h are shown in Figure 10. The results showed that with the increase of temperature, the apparent porosity of the specimens prepared with different silicon sources gradually decreased, while the density and compressive strength increased significantly. When the temperature was 1300 °C, the apparent porosity of porous mullite ceramic specimens prepared from fly ash, silica dust, and gangue were 8.74%, 34.83%, and 33.13%, respectively. The densities were 1.95 g/cm^3^, 1.85 g/cm^3^, and 1.91 g/cm^3^, respectively. The compressive strengths were 123.28 MPa, 107.88 MPa, and 147.84 MPa, respectively. The apparent porosity of specimens with fly ash as the silicon source was the smallest, but the compressive strength did not grow expectedly. This was because the glass phase in the substrate mainly formed closed pores, while promoting firing and crystallization. The formation of needle-shaped mullite in the specimens with gangue as the silicon source reduced the apparent porosity and enhanced the mechanical strength of the porous mullite ceramic.

## 4. Conclusions

In this study, AD after hydrometallurgical treatment was used as an aluminum source, and the three silicon sources were introduced into the mullitization reaction, to study the effects of different silicon sources on the preparation of porous mullite from the AD. The main findings were as follows:

When the sintering temperature reached 1200 °C, all of the specimens with fly ash, silica dust, and gangue as silicon sources could form mullite phases, and the structures of mullite were prisms, random aggregates, and needle-shaped, respectively. Among them, the formation of needle-shaped mullite in the specimens with gangue as a silicon source reduced the apparent porosity and enhanced the mechanical strength of the porous mullite ceramics; when the temperature was 1300 °C, the apparent porosity, density, and compressive strength were 33.13%, 1.98 g/cm^3^ and 147.84 MPa, respectively. The average activation energies of the phase transition process of the AD mixed with fly ash, silica dust, and gangue were 984 kJ/mol, 1113 kJ/mol, and 741 kJ/mol, respectively.

## Figures and Tables

**Figure 1 materials-15-07212-f001:**
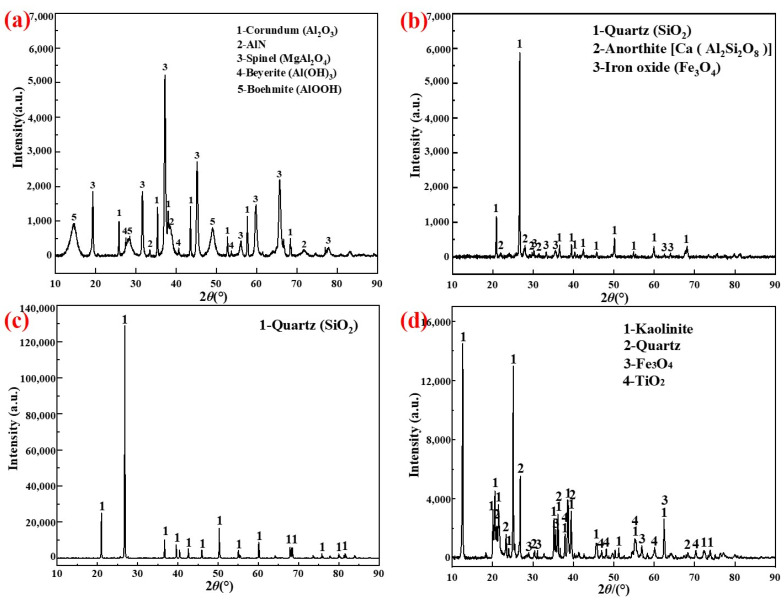
XRD of raw materials: (**a**) AD; (**b**) Fly ash; (**c**) Silica dust; (**d**) Gangue.

**Figure 2 materials-15-07212-f002:**
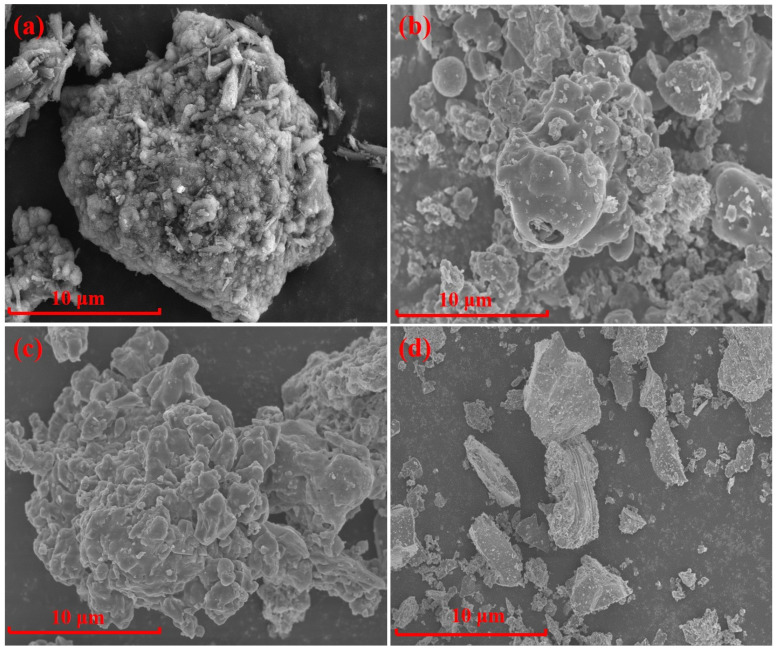
SEM of raw materials: (**a**) AD; (**b**) Fly ash; (**c**) Silica dust; (**d**) Gangue.

**Figure 3 materials-15-07212-f003:**
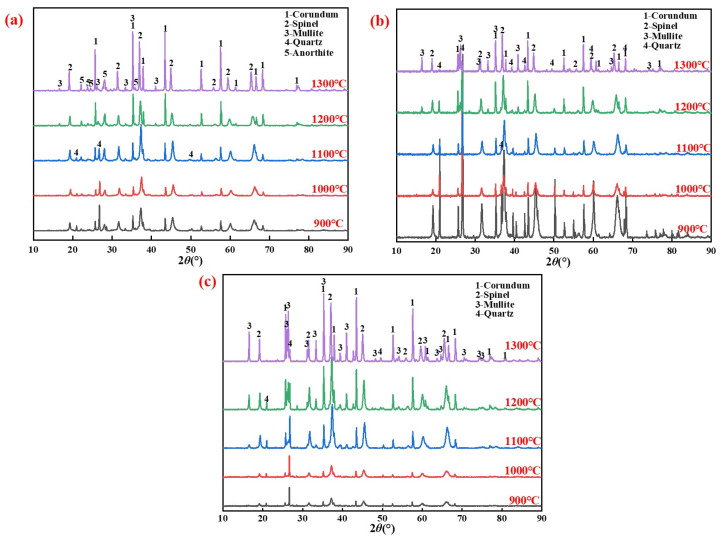
XRD patterns of specimens prepared from different silicon sources after sintering at 900–1300 °C: (**a**) Fly ash; (**b**) Silica dust; (**c**) Gangue.

**Figure 4 materials-15-07212-f004:**
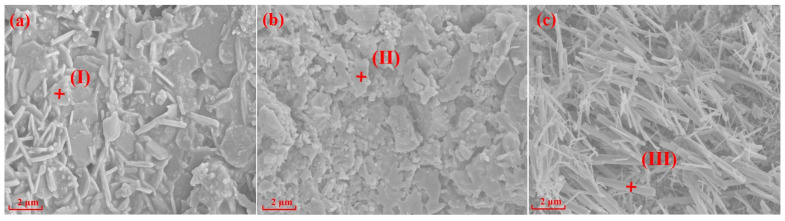
SEM and EDS of specimens with different silicon sources after sintering at 1200 °C: (**a**) Fly ash; (**b**) Silica dust; (**c**) Gangue.

**Figure 5 materials-15-07212-f005:**
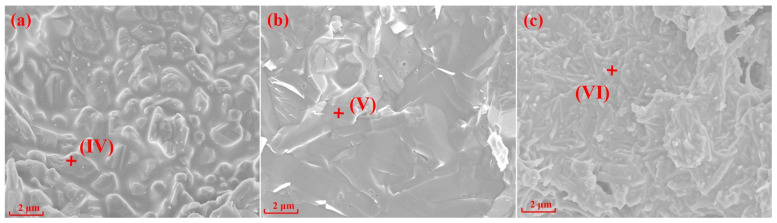
SEM and EDS of specimens with different silicon sources after sintering at 1300 °C: (**a**) Fly ash; (**b**) Silica dust; (**c**) Gangue.

**Figure 6 materials-15-07212-f006:**
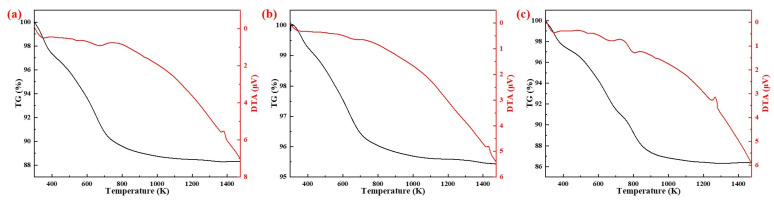
TG-DTA curves of samples from different silicon sources: (**a**) Fly ash; (**b**) Silica dust; (**c**) Gangue.

**Figure 7 materials-15-07212-f007:**
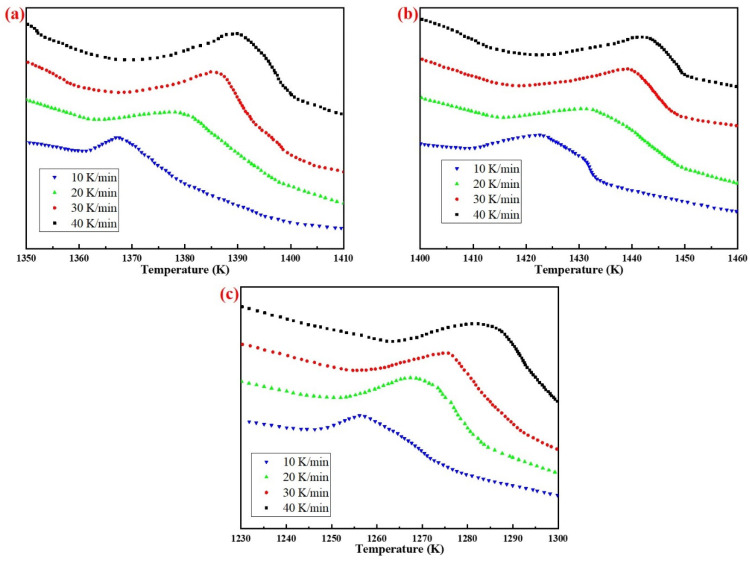
Differential thermal analysis of specimens from different silicon sources: (**a**) Fly ash; (**b**) Silica dust; (**c**) Gangue.

**Figure 8 materials-15-07212-f008:**
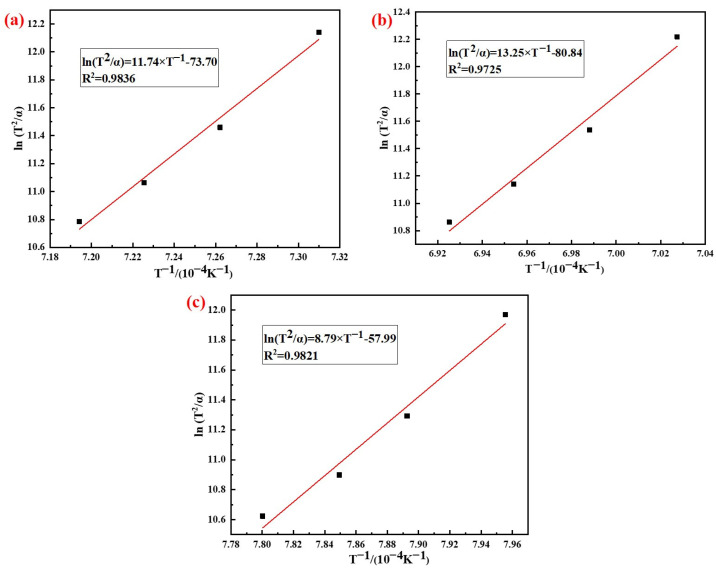
Analysis results of the Kissinger method: (**a**) Fly ash; (**b**) Silica dust; (**c**) Gangue.

**Figure 9 materials-15-07212-f009:**
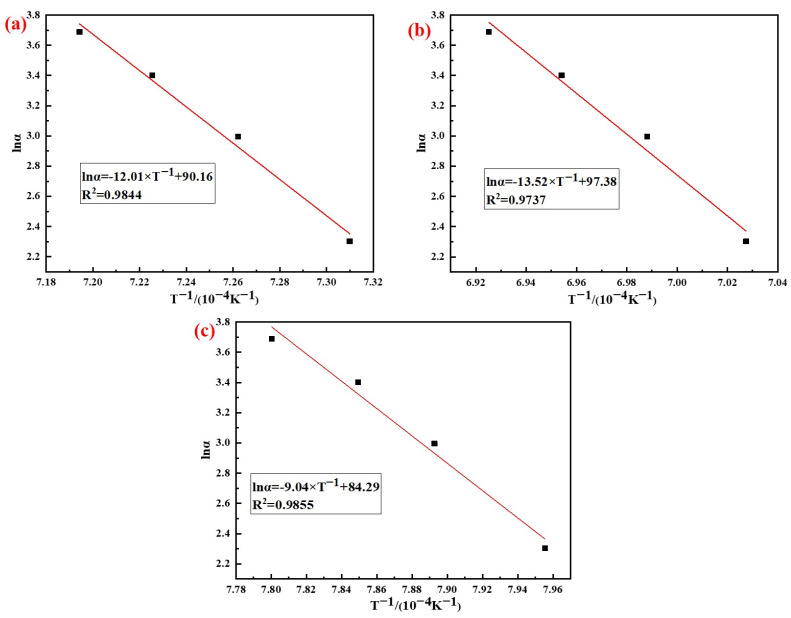
Analysis results of Ozawa method: (**a**) Fly ash; (**b**) Silica dust; (**c**) Gangue.

**Figure 10 materials-15-07212-f010:**
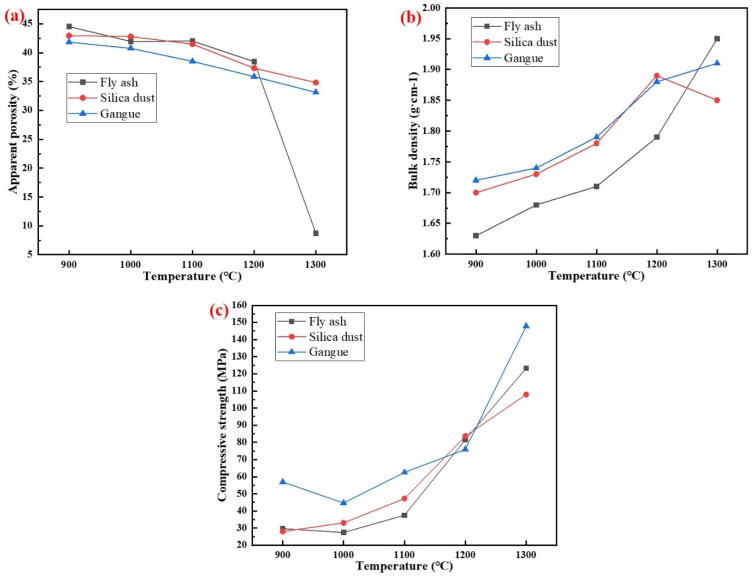
Physical properties of specimens with different silicon sources after sintering at 900–1300 °C for 2 h: (**a**) Fly ash; (**b**) Silica dust; (**c**) Gangue.

**Table 1 materials-15-07212-t001:** Chemical composition of the raw materials (wt%).

	Al_2_O_3_	SiO_2_	MgO	CaO	Fe_2_O_3_	TiO_2_	K_2_O	Else
**AD**	87.80	1.74	8.44	0.55	0.39	0.41	0.12	0.55
**Fly ash**	12.81	71.18	0.62	7.71	4.73	0.59	0.88	1.48
**Silica dust**	1.83	93.26	0.31	1.11	1.89	0.24	0.57	0.79
**Gangue**	35.03	59.08	0.44	0.77	1.31	2.27	0.65	0.45

**Table 2 materials-15-07212-t002:** The ratio of different samples (mass, g).

	AD	Fly Ash	Silica Dust	Gangue
**1**	6.7	3.3		
**2**	7.8		2.2	
**3**	6.8			3.2

**Table 3 materials-15-07212-t003:** EDS analysis.

Point	Elements and Content/wt%			
	Si	Al	O	Mg
**I**	7.0	38.2	53.5	1.3
**II**	20.6	28.3	51.1	—
**III**	12.9	30.3	56.8	—
**IV**	14.2	26.4	53.3	6.1
**V**	8.5	34.7	56.8	—
**VI**	11.3	43.0	41.7	4.0

**Table 4 materials-15-07212-t004:** Phase transition activation energies of specimens from different silicon sources.

Silicon Source	Activation Energy/kJ·mol^−1^		Average/kJ·mol^−1^
	Kissinger	Ozawa	
**Fly ash**	970	998	984
**Silica dust**	1102	1124	1113
**Gangue**	731	751	741

## Data Availability

The data presented in this study are available on request from the corresponding author. The data are not publicly available, due to industrial applications and confidentiality.

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
