# Peer review of "Effect of Silicon Source (Fly Ash, Silica Dust, Gangue) on the Preparation of Porous Mullite Ceramics from Aluminum Dross"

_materials, 2022, doi:10.3390/ma15207212_

Round 1

Reviewer 1 Report

Dear Authors,

basically an excellent paper. Very interesting and detailed machanistic information on heat treatment conditions for manufacturing of muliite from different industrial waste materials.

The only critizism concerns the quality of the images and tables. It would be beneficial if you could provide some higher resolution imges for Fig. 1 (acceptable but a bit blurry).

in Fig 3 the legend of the three sub-graphs is poorly readable uses larger font size.

Figure 4 and 5. The inserted EDS graphs - especially the element names - are unreadable, their content is shown in Table 3 anyway so I would recommend to remove the red insert (EDS spectra) in the graphs, this would also improve the quality of the graphs as such. Also change the corresponding text referring to the inserts in the graphs.

Table 2 I doubt that the accuracy of the measurements is high enough to give the numbers in four significant digits , three are enough (e.g 38.1 insted of 39.15)

the same is true for the activation energy data plotted in Table 3 give avctivation energies in integer numbers (without the decimal places, e.g. 987 instead of 987.23 KJ/mol)

One last point remamber that materials is WYSIWYG (what you see is what you get, so please check the positioning of the tables graphs sub and superscripts.

Author Response

Thank you for your  comments concerning our manuscript entitled “Effect of silicon source on the preparation of porous mullite ceramics from aluminum dross”, manuscript number materials-1957445. We have checked the referee’s comments very carefully and modified the paper substantially according to the referee’s review. Revised portion have been highlighted in red with the original text.

The only critizism concerns the quality of the images and tables. It would be beneficial if you could provide some higher resolution imges for Fig. 1 (acceptable but a bit blurry).

Thanks for your suggestion, we've change the images in Fig. 1.

in Fig 3 the legend of the three sub-graphs is poorly readable uses larger font size.

Thanks for your suggestion, we've change the images in Fig. 3.

Figure 4 and 5. The inserted EDS graphs - especially the element names - are unreadable, their content is shown in Table 3 anyway so I would recommend to remove the red insert (EDS spectra) in the graphs, this would also improve the quality of the graphs as such. Also change the corresponding text referring to the inserts in the graphs.

Thanks a lot for your remark to this point, we've remove the red insert (EDS spectra) in the graphs, and modified the text of the article.

Table 2 I doubt that the accuracy of the measurements is high enough to give the numbers in four significant digits , three are enough (e.g 38.1 insted of 39.15)

the same is true for the activation energy data plotted in Table 3 give avctivation energies in integer numbers (without the decimal places, e.g. 987 instead of 987.23 KJ/mol)

Thanks for your suggestion, We have reserved three significant digits for the measured data

Reviewer 2 Report

This article examines the formation of mullite by using aluminum dross waste (ADW) and different silica sources (fly ash, silica dust and gangue) as a substitution for clay. The research shows a high potential especially in recycling aluminum dross waste and silica for the production of Al2O3-SiO2 ceramics, which is a novel approach. However, with the publication, a few questions arose, that need minor revisions:

1.      Section 2.1: Would you please have a look at the D50 of your raw materials AD, fly ash, and silica dust? AD and the fly ash have the same D50. Is this a copy mistake? From my experience fly ash is much finer (also see your SEM images in Fig. 2).

2.      Section 2.2: the unit of heating rate is K/min.

3.      Section 3.1: Please check, if the different morphology of your mullite results in a different composition. There are two types of mullite: (3 Al2O3 · 2 SiO2) with 72 % Al2O3 and (2 Al2O3 · 1 SiO2) with 78 % Al2O3. The primary mullite (A3S2) is flaky and the secondary mullite (A2S) is needle-shaped (formation at a higher temperature). Especially in the case of the latter please use this term instead of “whiskers”.

4.      And please try to calculate the amount (%) of mullite (Rietveld refinement) which was formed with the different raw materials after sintering at 1300 °C.

Author Response

Thank you for your  comments concerning our manuscript entitled “Effect of silicon source on the preparation of porous mullite ceramics from aluminum dross”, manuscript number materials-1957445. We have checked the referee’s comments very carefully and modified the paper substantially according to the referee’s review. Revised portion have been highlighted in red with the original text.

  1. Section 2.1: Would you please have a look at the D50 of your raw materials AD, fly ash, and silica dust? AD and the fly ash have the same D50. Is this a copy mistake? From my experience fly ash is much finer (also see your SEM images in Fig. 2).

Thanks a lot for your remark to this point. The D50 of the AD is wrong, we have modified it. It is mentioned in the paper that the AD used in our research has been treated by hydrometallurgical treatment, so its particle size is smaller than the fly ash.

  1. Section 2.2: the unit of heating rate is K/min.

Thanks for your suggestion, we've change the ℃/min to K/min

  1. Section 3.1: Please check, if the different morphology of your mullite results in a different composition. There are two types of mullite: (3 Al2O3 · 2 SiO2) with 72 % Al2O3 and (2 Al2O3 · 1 SiO2) with 78 % Al2O3. The primary mullite (A3S2) is flaky and the secondary mullite (A2S) is needle-shaped (formation at a higher temperature). Especially in the case of the latter please use this term instead of “whiskers”.

Thank you for your valuable advice. According to our analysis of mullite samples prepared with different silicon sources and the research on the preparation of mullite in existing literatures, the different composition does affect the crystal morphology of mullite formed, especially the behavior of small amount of impurities contained in solid waste. In the next step, we will conduct further research according to your suggestion. In addition, we have changed the term "whiskers" to "needle shaped" in the text.

  1. And please try to calculate the amount (%) of mullite (Rietveld refinement) which was formed with the different raw materials after sintering at 1300 °C.

Thanks a lot for your remark to this point. We have attempted to measure the mullite content in samples accurately, including Rietveld refinement and RIR method. However, impurities in solid waste may cause some errors due to disorderly diffraction peaks formed after sintering. In the next step, we will try more methods to make the research more in-depth.

Reviewer 3 Report

Manuscript ID: materials-1957445

Title: Effect of silicon source on the preparation of porous mullite ceramics from aluminum dross

Authors: Hongliang Yan et al.

The title must be changed to “Effect of silicon source (fly ash, silica dust, gangue) on the preparation of porous mullite ceramics from aluminum dross”

Introduction.

Line 29. Authors must describe in detail the “economic and environmental problems” from aluminum dross.

Authors must write the novelty of this research compared to the previous studies. For example, why authors used fly ash, silica dust and gangue for mullite production?

Experimental.

Line 63. Authors used coal fly ash or another ash?

Table 1. What is the LOI (loss on ignition) content in samples? What is the carbon content (wt. %) in Fly Ash?

Figure 1, 3. Authors must write the phase names of each XRD legend.

Figure 1d. Authors must sign all XRD peaks.

Section 2.2. What is the atmosphere during sintering experiments? Write the sample ratios in g in a separate table.

Section 3.3. Author must add information how points from Figures 7-8 were calculated. Add more information to TGA curves. Not all scientists know how kinetics are calculated from TGA curves. An explanation needs to be made.

Figure 9. The authors obtained data on the physical properties of the resulting ceramics. It is necessary to compare the obtained data with the previously obtained results of other researchers and with samples of mullite ceramics, which is currently used in industry.

References. Authors can add more new links (5-7 sources) about mullite ceramics production from solid wastes from 2020-2022 years.

Technical errors:

The references (references in text and references list) didn’t write in Materials style. Read more about it here: https://www.mdpi.com/journal/materials/instructions

Line 155. Use “Ea” instead of “E” in all article text.

Author Response

The title must be changed to “Effect of silicon source (fly ash, silica dust, gangue) on the preparation of porous mullite ceramics from aluminum dross”

Thanks for your suggestion, we've change the title

Introduction.

Line 29. Authors must describe in detail the “economic and environmental problems” from aluminum dross.

Thanks for your suggestion, we've add some details of the “economic and environmental problems” from aluminum dross and make corresponding changes to the text. The “economic and environmental problems” from aluminum dross are: (1) AlN in AD will hydrolyze with water even at room temperature, producing NH3 and polluting the environment; (2) The accumulation of AD will cause the leaching of salts (KCl, NaCl, NaF, et al) and damage the soil environment gradually (3) The valuable secondary resources in AD have not been utilized effectively.

Authors must write the novelty of this research compared to the previous studies. For example, why authors used fly ash, silica dust and gangue for mullite production?

Thanks for your suggestion,We’ve added the novelty of this research compared to the previous studies in section 1.

Experimental.

Line 63. Authors used coal fly ash or another ash?

Thanks a lot for your remark to this point, the raw material is fly ash.

Table 1. What is the LOI (loss on ignition) content in samples? What is the carbon content (wt. %) in Fly Ash?

Thanks for your suggestion.The LOI (loss on ignition) content of each samples can be obtained from the results of TG-DTA analysis in Fig. 6 (Supplementary). The carbon content (wt. %) in Fly Ash was not analyzed, The main component of the raw materials were measured by XRF only.

Figure 1, 3. Authors must write the phase names of each XRD legend.

Thanks a lot for your remark to this point, we've add the phase names of each XRD legend in Fig 1 and Fig 3.

Figure 1d. Authors must sign all XRD peaks.

Thanks a lot for your remark to this point, we've sign all XRD peaks in Fig 1.

Section 2.2. What is the atmosphere during sintering experiments? Write the sample ratios in g in a separate table.

Thanks for your suggestion, the atmosphere during sintering experiments was air atmosphere, We have made corresponding modifications in the paper and added a table of sample ratio in Section 2.2.

Section 3.3. Author must add information how points from Figures 7-8 were calculated. Add more information to TGA curves. Not all scientists know how kinetics are calculated from TGA curves. An explanation needs to be made.

Thanks for your suggestion. Figures 7-8 were calculated from Equation 5 and Equation 6. We’ve add the Fig 6 (TG-DTA curves of samples from different silicon sources) which could explains the origin of the TGA curve

Figure 9. The authors obtained data on the physical properties of the resulting ceramics. It is necessary to compare the obtained data with the previously obtained results of other researchers and with samples of mullite ceramics, which is currently used in industry.

Thanks for your suggestion. We studied the effect of different silicon sources on the preparation of mullite, The physical properties of different silicon sources(fly ash, silica dust, gangue) were compared. That’s why we didn’t compared the obtained data with the previously obtained results of other researchers or with samples of mullite ceramics, which is currently used in industry.

References. Authors can add more new links (5-7 sources) about mullite ceramics production from solid wastes from 2020-2022 years.

Thanks for your suggestion. We've add some new links about mullite ceramics production from solid wastes from 2020-2022 years.

Technical errors:

The references (references in text and references list) didn’t write in Materials style. Read more about it here: https://www.mdpi.com/journal/materials/instructions

Line 155. Use “Ea” instead of “E” in all article text.

Thanks a lot for your remark to this point, we've change the“E” to “Ea”.

Round 2

Reviewer 3 Report

Article can be accepted in this form.